# Unruptured Anterior Communicating Artery Aneurysms: Management Strategy and Results of a Single-Center Experience

**DOI:** 10.3390/jcm12144619

**Published:** 2023-07-11

**Authors:** Katarzyna Wójtowicz, Lukasz Przepiorka, Sławomir Kujawski, Andrzej Marchel, Przemysław Kunert

**Affiliations:** 1Department of Neurosurgery, Medical University of Warsaw, 02-091 Warsaw, Poland; 2Department of Exercise Physiology and Functional Anatomy, Ludwik Rydygier Collegium Medicum in Bydgoszcz, Nicolaus Copernicus University in Toruń, 85-077 Bydgoszcz, Poland

**Keywords:** unruptured intracranial aneurysms, microsurgery, endovascular, observation, management, anterior communicating artery, morbidity, mortality

## Abstract

Although anterior communicating artery (AComA) unruptured intracranial aneurysms (UIAs) comprise one of the largest aneurysm subgroups, their complex adjacent neurovasculature and increased risk of rupture impede optimal management. In the present study, we analyzed the results of our diverse strategy in AComA UIAs with the additional goal of assessing the risk of treatment and the incidence of hemorrhage. We analyzed 131 patients, of which each was assessed by a multidisciplinary neurovascular team and assigned to observation (45.8%), endovascular treatment (34.4%) or microsurgery (19.8%). Median aneurysm sizes were 3, 7.2 and 7.75 mm, respectively. In the observation group, four (7.1%) aneurysms (initially <5 mm) grew over a median time of 63.5 months and were treated endovascularly. We found that fewer patients in the observation group were smokers (*p* = 0.021). The aneurysm size ratio was different between the combined treatment versus the observation group (*p* < 0.0001). Noteworthily, there were no hemorrhages in the observational group. Mortality for all patients with available follow-up was 2.4% (3/124) and permanent morbidity was 1.6% (2/124) over a mean follow-up of 64.2 months. These compelling rates refer to a high-risk group with potentially devastating consequences in which we have decreased the annual risk of hemorrhage to 0.14%.

## 1. Introduction

The prevalence of unruptured intracranial aneurysms (UIAs) is estimated at 3.2% [1]. The anterior communicating artery (AComA) region stands out from other anatomical locations due to its congestion of functionally important structures, namely, numerous perforators, and relatively frequent prevalence of AComA UIAs. Additionally, AComA UIAs may exhibit twice the risk of rupture than other UIAs with an annual risk of rupture as high as 2.2% [2,3].

Even though AComA aneurysms comprise the biggest subgroup of all UIAs, their optimal management remains unclear. As with any other UIA, each patient needs to be qualified for observation or treatment. Endovascular techniques and microsurgery are standard methods of treatment, though the choice of which of them for patients qualified for treatment is often subjective.

To overcome this lack of unequivocal management guidelines, numerous scales and algorithms have been designed to facilitate the decision-making process [4,5,6]. In the present study, we analyze the results of our diverse management strategy in patients with AComA UIAs. We additionally set out to assess the risk of treatment including complications, the immediate and long-term outcomes of the endovascular treatment as well as open surgery. Moreover, we endeavored to measure the incidence of aneurysmal hemorrhage in each of three management option groups.

## 2. Materials and Methods

This study was designed as a retrospective, single-center, consecutive case series, undertaken in an academic setting from 2011 to 2021. For each patient, the multidisciplinary team—comprising neurological surgeons and neurointerventional radiologists—qualified patients for observation or treatment. Our management strategy changed over time. At the beginning, it was based on the algorithm published by Chalouhi et al. [7]. Over time, we gradually remodeled the assessment of the risk of hemorrhage according to the PHASES and integrated additional tools as well as the UIATS recommendations [4,5]. When needed, the multidisciplinary team chose the treatment modality. In general, endovascular treatment was the preferred modality for patients qualified for treatment, unless the aneurysm configuration made it particularly hazardous, in which case microsurgery was chosen.

We analyzed our long-term clinical follow-up for all groups. In the observational and endovascular groups, we analyzed radiological outcomes, whereas in the endovascular and microsurgical groups, we analyzed clinical outcomes immediately and at discharge. Clinical outcomes were assessed with the modified Rankin scale (mRS). Good outcomes were defined as an mRS score of 0 to 2, and poor outcomes were defined as an mRS score greater than 2. Neurological deficits were divided into minor (mRS 1 to 2) and major (mRS greater than 2). Deterioration was defined as an increase of at least 1 point in the mRS score. Follow-up information was obtained during routine clinic visits or telephone interviews.

### 2.1. Follow-Up Algorithm

Patients in the observational group were monitored as follows: outpatient clinic visits with a computed tomography angiography (CTA) or magnetic resonance angiography (MRA) at six months and then once a year thereafter.

In the endovascular group, we routinely perform a posttreatment digital subtraction angiography (DSA) approximately 6 months after the endovascular procedure, and in case of a satisfactory result, this is the last radiological study performed. Subsequent DSAs are performed only when necessary (i.e., incomplete occlusion in the first posttreatment DSA, visible remnant, etc.). We use the Raymond–Roy (RROC) grading scale to assess aneurysm occlusion [8]. As a rule in our institution, we waive routine postoperative vascular imaging (DSA, CTA, and MRA) after microsurgery. From 2018, we started routinely using intraoperative indocyanine green videoangiography to confirm the patency of the arteries and closure of the aneurysm. Prior to that, we performed early postoperative CTA in uncertain cases.

### 2.2. Statistical Analyses

We performed the Shapiro–Wilk and Levene’s tests to examine assumptions of data normality and the equality of variances, respectively. We examined the associations between qualitative variables with the chi-squared or Fisher’s exact tests. Differences between two groups were evaluated with the Mann–Whitney U or independent *t*-tests, depending on the assumptions met. Differences between more than two groups were evaluated with the Kruskal–Wallis H test or one-way ANOVA, depending on the assumptions met. Post hoc comparisons were made using the Dunn test, and values were adjusted for multiple comparisons. All analyses were performed using R [9], and violin graphs were created with the ggstatsplot library [10]. Effect size (ε) and confidence interval [−95%; 95%] from the ggstatsplot library are reported for the Kruskal–Wallis test result. All analyses were performed with a significance level of α = 0.05.

## 3. Results

During the study period 131 patients were diagnosed with 131 UIAs AComA. Initially, 60 cases (45.8%) were classified to the observation group and 71 cases (54.2%) to the treatment group, of which, 45 cases (63.4%) were qualified to endovascular and 26 cases (36.6%) to microsurgical treatment (Table 1 and Figure 1). After crossovers, we analyzed 64 patients in the observational group, 44 patients in the endovascular treatment group and 25 patients in the microsurgery group. These numbers exceed the total number of patients diagnosed with UIAs AComA. The reason for that is that patients initially observed—until diagnosed with aneurysmal growth—were included in both the observation group (for the whole observation period) and the endovascular group (after qualification for treatment). The demographics and aneurysmal risk factors for each analyzed group are presented in Table 2.

### 3.1. Observation Group

#### 3.1.1. Assignment

Sixty patients were initially qualified for the conservative management specified earlier. Additionally, four patients initially assigned to the endovascular treatment group were moved to the observational group. Three patients refused endovascular treatment. In one case during diagnostic DSA, the risk of an endovascular treatment had been assessed as high, and treatment was not undertaken. For that reason, the analyzed observational group comprised 64 subjects. Significantly fewer patients in the observation group were smokers (chi-square test, *p* = 0.021). The aneurysm size ratio was significantly different between the combined treatment groups versus the observation group (Kruskal–Wallis *p* < 0.0001, ε = 0.45 [0.33; 1]). Post hoc tests showed that the size ratio was significantly smaller in the observational group in comparison to the endovascular and microsurgery groups (both *p* < 0.0001) (Figure 2A).

In addition, the aneurysm aspect ratio was significantly different between examined groups (Kruskal–Wallis *p* < 0.0001, ε = 0,31 [0.19; 1]). Post hoc tests showed that the aspect ratio was significantly higher in the endovascular and microsurgery groups than in the observational group (*p* < 0.0001 and *p* = 0.0002, respectively) (Figure 2B).

Subsequently, during the observation period, in four cases (6.25%), we observed aneurysmal growth, and these patients were qualified for endovascular treatment. These aneurysms were not significantly different from the remaining aneurysms in statistical analysis. Of note is that these aneurysms—up till their growth—were included in the observation group analysis. In the observation group, three patients (4.67%) had poor initial neurological condition (mRS 3 or more). These were caused by unrelated conditions such as ischemic stroke (n = 2) and lower extremity peripheral artery disease (n = 1).

#### 3.1.2. Radiological Outcomes

Radiological follow-up data were available for 56 patients (87.5%). During the observation period, four (7.1%) aneurysms grew and were treated endovascularly. The median time to the aneurysm growth was 63.5 months (range: 54–73 months); in each case, the initial aneurysm size was under 5 mm. For the remaining (in other words, stable) 52 (92.9%) aneurysms, the mean radiological follow-up was 60.7 months (range: 5–184). The median follow-up for aneurysms under observation, including aneurysms that eventually grew, was 60.9 (range: 5–184).

#### 3.1.3. Clinical Outcomes

Clinical outcomes were available for 63 patients (98.4%); however, 1 patient was lost to the long-term follow-up. Mean clinical follow-up was 71.5 months (range: 7–187). There were no subarachnoid hemorrhages (SAH) or other symptoms of UIA during this period. Five patients (7.9%) died because of unrelated reasons.

### 3.2. Endovascular Treatment

#### 3.2.1. Assignment

The endovascular group primarily included 45 patients. Of these, one patient (2.2%) chose treatment in another institution, and three patients (6.67%) refused endovascular treatment. In the remaining case (2.2%), after a diagnostic DSA, due to the expected high risk of treatment, an endovascular procedure was not initiated. Moreover, four patients (6.7%) from the observational group were diagnosed with aneurysmal growth and qualified for treatment. No significant differences were found in the size, aspect ratio and size ratio between those four patients and the rest of the endovascular group (all *p* > 0.05). Finally, after relocations, 44 patients were treated endovascularly.

The following techniques were used: coiling alone in 59.1% of cases, coiling with stent in 20.5%, coiling with flow-diverting devices in 11.4% and flow-diverting alone in 9.1% of cases. One case (2.3%) required two-staged treatment.

In total, 45 endovascular procedures were performed to treat 44 aneurysms. The intraprocedural complication rate was 17.8% (eight cases), of which 75% (six cases) were asymptomatic. Coil protrusion to the artery occurred in 8.9% of cases (n = 4), after which, stent deployment was required. The rate of vasospasm was 2.2%. The intraprocedural rupture rate for all endovascular procedures was 6.7% (three cases).

#### 3.2.2. Clinical Outcomes

On the first postprocedural day, we observed deterioration in five cases (11.1%). In three of these cases, these were minor; in the other two cases, there were major deficits. Deteriorations were caused by ischemic stroke in two cases, which was confirmed by diffusion-weighted imaging (DWI) magnetic resonance (MR). In two other cases, worsening was caused by an intraprocedural hemorrhage (one was an aneurysmal intraoperative rupture, and the second, more serious, was due to an injury of the A2 segment of anterior cerebral artery). A patient with a major deficit due to an intraprocedural hemorrhage and anterior cerebral artery injury died during hospitalization. There were no intraprocedural complications or ischemia in the postoperative MR in the remaining one patient.

At their discharge from hospital, the neurological status of 41 patients (93.2%) was stable, with their mRS scores as before the treatment; 2 patients (4.5%) were discharged in a worse neurological status due to a new, minor deficit in each case. At discharge from hospital, procedure-related permanent morbidity and mortality rates were 4.5% and 2.3%, respectively. Clinical follow-up data were lost for one patient. Four patients died: one during hospitalization, one due to hemorrhage from the treated aneurysm 57 months after treatment and two due to unrelated reasons. There was one hemorrhage for 186.3 patient years.

#### 3.2.3. Radiological Outcomes

Complete obliteration at the time of treatment was achieved for 38 (86.4%) aneurysms. One patient died shortly after the treatment (see above) and three patients were lost to follow-up. For the remaining 40 patients, the median radiological follow-up was 7.25 months (range: 3–14). The complete occlusion rate was 90%, and the near-complete occlusion (visible neck remnant) rate was 5%, while the dome remnant rate was 5%. Of the two aneurysms with dome remnants, one required retreatment, and the other was left untreated and remained under observation due to a high risk of retreatment. Aneurysms with neck remnants (5%) remained under further observation.

In the statistical analysis, aneurysms with near-complete occlusion and aneurysms with dome remnants were significantly bigger (aneurysm size, mm (median (interquartile range)): 10.9 (2.97) vs. 6.72 (3.09), *p* = 0.009; size ratio > 3:4.66 (1.72) vs. 2.83 (1.33), *p* = 0.01; no significant differences in aspect ratio > 1.6) as compared to the completely occluded aneurysms.

Ultimately, 23 patients (52%) patients had a repeated long-term DSA in a mean follow-up of 37.6 months. Recanalization of four previously occluded aneurysms (17.4%) was demonstrated. These patients exhibited more frequent hypertension (Fisher’s exact test *p* = 0.03) and multiple aneurysms (Fisher’s exact test *p* = 0.04). In long-term follow-up, two aneurysms with neck remnants had complete occlusion without additional treatment. Additionally, complete occlusion was achieved in the retreatment case.

### 3.3. Microsurgical Group

#### 3.3.1. Assignment

Initially, 26 patients (19.8%) were qualified for elective microsurgery. Of these, 1 patient chose a different institution for treatment, hence, in the final assignment of microsurgery, there were 25 patients. When compared with other groups in our study, these aneurysms caused the mass effect significantly more often (Fisher’s exact test *p* = 0.04).

#### 3.3.2. Postoperative Complications

On the postoperative day one, deterioration was noted in seven patients (28%). In one case, the postoperative deficit was temporary. Deteriorations were caused by an ischemic stroke recognizable in CT in six cases (24%). Additionally, one of these patients (4%) had a postoperative epidural hematoma. In the remaining one case (4%), the cause of neurological deterioration was not found. Median aneurysmal size in patients who deteriorated was 7 mm (range: 5–13 mm).

#### 3.3.3. Clinical Outcomes

At discharge from hospital, 19 patients (76%) were stable with their mRS scores as before the treatment. Almost a quarter of patients deteriorated, including five minor deficits (20%) and one major deficit (4%). The patient with a major deficit died one month after the surgery due to medical complications that were a consequence of a neurosurgical treatment.

Long-term clinical follow-up was available for 21 patients (84%), while 4 patients (16%) were lost to long-term follow-up. Mean follow-up was 63 months (range 1–123). Out of six patients (24%) with postoperative deterioration, three patients improved, one patient died (see above), one patient was lost to the follow-up and one patient exhibited permanent deterioration. There was no hemorrhage from the treated aneurysms. Overall mortality for this group was 4.8%, and permanent morbidity (mRS 1) was 4.8%.

### 3.4. Overall Results

Mortality for all patients (observed, treated endovascularly and treated microsurgically) with available follow-up was 2.4% (3/124), and permanent morbidity was 1.6% (2/124) over a mean follow-up of 64.2 months.

## 4. Discussion

In this study we presented our management strategy with AComA UIAs and analyzed their treatment results. We focused exclusively on AComA aneurysms because their increased risk of rupture makes them distinct from other UIAs [11]. Management of patients with AComA UIAs still poses a considerable challenge. Treatment of UIAs is intended to prevent SAH, but it involves a substantial risk of morbidity and mortality. At the beginning, in our decision-making process, we followed the algorithm published by Chalouhi et al. [7]. Subsequently, we introduced additional tools: PHASES and the UIATS system score. Applying these scales did not simplify the decision-making process; their recommendations often differed. Additionally, in 43.5% of cases, UIATS suggestions were not definitive. According to the most recent European recommendations, the indications for treatment are even more limited [12]. Even though our management was tailored for each individual case, some general trends emerged and will be discussed below.

### 4.1. Observational Group

Morphological features of aneurysms, such as their size, size ratio and aspect ratio, were significantly smaller in the observational group. Conversely, patients in the treatment group were significantly more often smokers. As a result of our strategy, there were no intracranial hemorrhages or neurological deteriorations caused by aneurysm in the observational group.

### 4.2. Endovascular Group

In the endovascular group, procedure-related mortality (that was a consequence of the provided treatment) and permanent morbidity were 2.3%. Morbidity in our series is lower, while the mortality is comparable to that from the meta-analysis of the endovascular treatment of AComA aneurysms by Fang et al. in 2014 [13]. In that study, permanent morbidity was 8% (95% CI 3–20%) for AComA UIAs, and the procedure-related mortality was 2% (95% CI 1–9%) for AComA UIAs. The rate of hemorrhage was 2.3% in our series and is comparable to the meta-analysis results by Fang et al. (2%, 95% CI 0–6%) [13]. In that study, the retreatment rate was 3%, compared to 5% in our series.

In our study, the long-term complete and near-complete occlusion rates were 90% and 95%, respectively, which is similar to what Fang et al. found in their meta-analysis, i.e., 90% for both complete and near-complete occlusions.

A systematic analysis by O’Neill et al. presents data separately for coiling and stent-assisted coiling. Good clinical outcomes were achieved in 99.2% and 92.1% for these two treatment modalities, while treatment-related mortality was 0 and 1.1%, respectively [14]. There were no SAHs from the treated aneurysms in either endovascular treatment technique. Retreatment rates were 4.9% and 6.8%, for coiling and stent-assisted coiling, respectively. All these results are similar to those obtained in our series.

### 4.3. Microsurgical Group

Morbidity and mortality in our microsurgical group were 4.7% and 4.7% respectively. In the review of the literature by Nussbaum, 3.3% of microsurgically treated patients did not achieve a good clinical outcome, while the mortality rate was 0.3% [15]. The mortality rate in our study is higher (4.7%), but it refers to a single patient in a comparatively small group (n = 25). This patient had an aneurysm that was not amenable to endovascular techniques and had a high risk of rupture. Similarly, the majority of microsurgically treated aneurysms had complex anatomy, and endovascular treatment of these aneurysms was evaluated as too risky.

The higher risk of clipping may be related to a decreasing volume of patients treated microsurgically, which imperils the opportunity to develop, improve and sustain high operative abilities.

### 4.4. Final Comments

The observation of aneurysms did not expose patients to hemorrhage and at the same time avoided the risk of treatment, possibly reducing the mortality and morbidity in the whole group. On the other hand, despite the risk of morbidity of endovascular and microsurgical treatment in our material being 2.3% and 4.7%, respectively, the cumulative morbidity risk for the whole study population was lower, at 1.6%. Ultimately, it appears beneficial to observe AComA UIAs whenever reasonable and—when treatment is warranted—diversify preventive aneurysm repair modalities.

In a meta-analysis of the natural history of AComA UIAs, Mira et al. estimated the annual risk of rupture at 2.2% [2]. Yet, in our study, with 690.15 patient years of follow-up (129 patients and 5.35 years) and one case of SAH, we managed to decrease this number to the 0.14% annual risk of rupture.

Based on our experience, we conjecture that evaluating each AComA UIA in a multidisciplinary team, with a preference to observe whenever possible and reasonable, may be of benefit. Prior to such evaluation a patient should be carefully evaluated using all available UIA scales (PHASES, UIATS, and ISUIA).

### 4.5. Limitations

Our study is limited by its retrospective nature and small sample size. Additionally, our multidisciplinary team evaluates many patients from outside hospitals. Most of these patients, when qualified for observation, are not followed in our institution, and thus, such patients were not included in our database. As a result, there is an inflated portion of treated patients, relative to the whole study group, whereas the patients in the observational group constitute only a fraction of patients actually surveilled in our outpatient clinics.

Finally, our management strategy changed over time according to updated guidelines and risk calculators and is not homogenous for the evaluated patients.

## 5. Conclusions

Using diversified management of AComA UIAs, we have decreased the annual risk of SAH to 0.14% at the expense of 2.4% mortality and 1.6% permanent minor deficit rates. Noteworthily, there were no hemorrhages in the observational group. The morbidity and mortality in AComA UIAs are compelling, but these refer to a high-risk group with potentially devastating consequences.

## Figures and Tables

**Figure 1 jcm-12-04619-f001:**
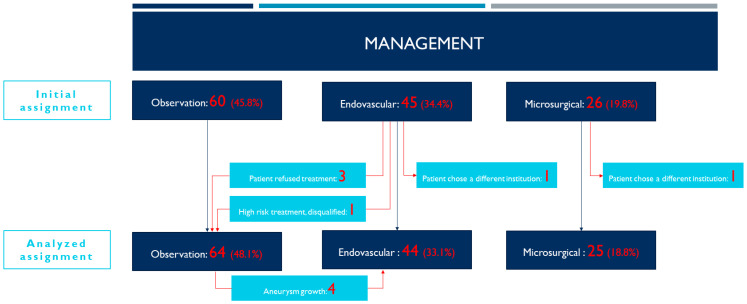
Management of patients with unruptured anterior communicating artery aneurysms as they were initially assigned and then analyzed in our study.

**Figure 2 jcm-12-04619-f002:**
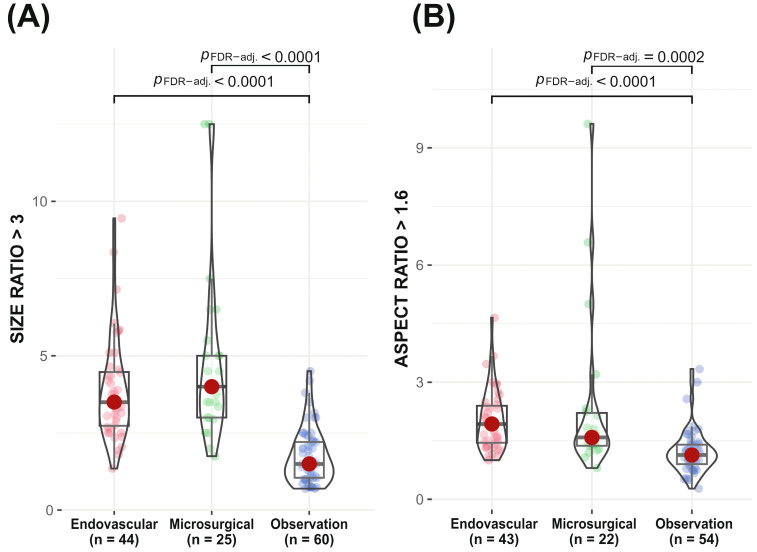
Comparison of size ratio (**A**) and aspect ratio (**B**) between endovascular, microsurgical and observation groups. Red dots and horizontal black lines inside the box indicate median value. Magenta dots in endovascular, green dots in microsurgical and blue in observation groups denote the results of individual patients. The shape of the violin graph indicates the distribution of results.

**Table 1 jcm-12-04619-t001:** Demographics and aneurysmal characteristics of groups of patients as initially assigned. Abbreviations: IA—intracranial aneurysm, SAH—subarachnoid hemorrhage.

Data	Observation Group No. (%)	Endovascular Group No. (%)	Microsurgical Group No. (%)	*p*
Initial assignment	60 (45.8%)	45 (34.4%)	26 (19.8%)	
Sex				*p* = 0.8
male	23 (38.3%)	20 (44.44%)	10 (38.46%)	
female	37 (61.7%)	25 (55.56%)	16 (61.54%)	
Age, years				*p* = 0.1
median	65	62	57.5	
patients under 40 years old	5 (8.3%)	4 (8.8%)	3 (11.5%)	*p* = 0.9
Family history of SAH or IAs	5 (8.3%)	5 (11.1%)	4 (15.4%)	*p* = 0.8
Previous SAH from another aneurysm	5 (8.3%)	1 (2.2%)	1 (3.8%)	*p* = 0.42
Hypertension	38 (63.3%)	36 (80%)	23 (88.5%)	*p* = 0.06
Smoking	17 (28.3%)	20 (44.4%)	15 (57.7%)	*p* = 0.02
Initial medical condition (mRS)				
0	56 (93.3%)	43 (95.6%)	23 (92%)	
1	2 (3.3%)	1 (2.2%)	1 (4%)	
2	0	0	1 (4%)	
3	1 (1.7%)	1 (2.2%)	0	
4	1 (1.7%)	0	0	
5	0	0	0	
Aneurysm size, mm				*p* < 0.001
median	3	7.2	7.75	
range	1.4–9	2.7–18.9	3.5–25	
Aneurysm size groups				
<5 mm	49 (81.7%)	6 (13.3%)	3 (11.5%)	
5–7 mm	8 (13.3%)	15 (33.3%)	9 (34.6%)	
>7–12 mm	3 (5%)	20 (44.4%)	9 (34.6%)	
>12–25 mm	0	4 (8.9%)	3 (11.5%)	
>25 mm	0	0	2 (7.7%)	
Aspect ratio > 1.6	10 (16.7%)	27 (60%)	11 (42.3%)	*p* < 0.0001
Size ratio > 3	5 (8.3%)	24 (53.3%)	18 (69.2%)	*p* < 0.0001
Multilobulated	5 (8.3%)	17 (37.8%)	1 (3.8%)	*p* < 0.001
Multiple	22 (36.7%)	19 (42.2%)	10 (38.5%)	*p* < 0.9
Symptomatic aneurysms	0	0	2 (8%): 1 (4%) visual symptom and 1 (4%) seizure	*p* < 0.04

**Table 2 jcm-12-04619-t002:** Demographics and aneurysmal characteristics of analyzed groups of patients.

Data	Observation Group No. (%)	Endovascular Group No. (%)	Microsurgical Group No. (%)	*p*
Analyzed assignment *	64 (48.1%)	44 (33.1%)	25 (18.8%)	
Sex				*p* = 0.839
male	24 (37.5%)	19 (43.2%)	10 (40%)	
female	40 (62.5%)	25 (56.8%)	15 (60%)	
Age, years				*p* = 0.072
median	65.5	62	56	
patients under 40 years old	5 (7.8%)	4 (9%)	3 (12%)	
Family history of SAH or IAs	5 (7.8%)	5 (11.4%)	4 (16%)	*p* = 0.675
Previous SAH from another aneurysm	0	0	1 (4%)	*p* = 0.606
Hypertension	42 (65.6%)	34 (77.3%)	23 (92%)	*p* = 0.062
Smoking	17 (26.6%)	19 (43.2%)	14 (56%)	*p* = 0.024
Initial medical condition (mRS)				
0	59 (92.2%)	42 (95.5%)	23 (92%)	
1	2 (3.1%)	2 (4.5%)	1 (4%)	
2	0	0	1 (4%)	
3	2 (3.1%)	0	0	
4	1 (1.6%)	0	0	
5	0	0	0	
Aneurysm size, mm				*p* < 0.001
median	3	7	8	
range	1.4–9	2.7–18.9	3.5–25	
Aneurysm size groups				
<5 mm	49 (76.6%)	6 (13.6%)	2 (8%)	
5–7 mm	10 (15.6%)	16 (36.4%)	6 (36%)	
>7–12 mm	5 (7.8%)	19 (43.2%)	12 (48%)	
>12–25 mm	0	3 (6.8%)	3 (12%)	
>25 mm	0	0	2 (8%)	
Aspect ratio > 1.6	12 (18.75%)	27 (61.4%)	11 (44%)	*p* < 0.0001
Size ratio > 3	8 (12.5%)	22 (50%)	19 (76%)	*p* < 0.0001
Multilobulated	6 (9.4%)	18 (40.9%)	1 (4%)	*p* < 0.001
Multiple aneurysms	23 (35.9%)	20 (45.5%)	9 (36%)	*p* = 0.572
Symptomatic aneurysms	0	0	2 (8%): 1 (4%) visual symptom and 1 (4%) seizure	*p* = 0.034

* Sum of analyzed patients exceeds total number of patients—an explanation is provided in the text. Abbreviations: IA—intracranial aneurysm, SAH—subarachnoid hemorrhage.

## Data Availability

The data presented in this study are available on request from the corresponding author after acceptance of all the co-authors.

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
