# Peer review of "Unruptured Anterior Communicating Artery Aneurysms: Management Strategy and Results of a Single-Center Experience"

_jcm, 2023, doi:10.3390/jcm12144619_

Round 1

Reviewer 1 Report

In their article, the authors describe the treatment of unruptured ACOM aneurysms. Depending on the findings, the patients were observed, endovascularly treated or surgically treated.

In my opinion, the work does not currently show any innovations and merely confirms the data from the literature.
In addition, the 3 groups are controlled with different methods (CTA vs. DSA. vs. no regular follow-up during surgery). The follow-up of endovascular treatment (DSA) is very short with an average of 7 months. Only just under 50 % were monitored over 24 months.
Radiological follow-up in the observation group was only available in 87.5%. Thus, a statement is also only possible in these patients.
The difference in numbers in Figure 2 (A vs. B) cannot be explained.
The reduced rate of morbidity / mortality in observation is referred to a study from 2006. More recent studies were desirable.

Minor points: Table 1 and 2 are almost identical - one could be deleted. In Table 1, the age in the surgery group is missing.

Author Response

Response to Reviewer 1 Comments

In their article, the authors describe the treatment of unruptured ACOM aneurysms. Depending on the findings, the patients were observed, endovascularly treated or surgically treated.

Point 1: In my opinion, the work does not currently show any innovations and merely confirms the data from the literature.

Response 1: The Authors are grateful to the Reviewer for meticulous comments and in-depth analysis of our manuscript. We appreciate the criticism and would like to point out that the novelty of our work lies in the fact that it is the first to report on our strategy (which is in fact the first of its kind). Additionally, we would like to highlight that reports on all treatment modalities, including observation, are scarce, as most papers focus on a specific treatment method, e.g., flow diverters or clipping. We strongly believe that a broader approach, as reported here, is critically important. The following arguments further demonstrate the importance of presence of our results in scientific literature. Firstly, UIA recommendations change over time (which is a completely natural process), but each their iteration highlights the value of the personal, individualized approach to each patient and aneurysm. Secondly, numerous UIA evaluation treatment scales (now easily available even in the form of online calculators) commonly provide opposing recommendations for the same aneurysm. For that reason, decision making in UIAs is often arguable and controversial. To support managing patients, we provide real-world data of all treatment options.

An additional novelty lies in the fact that our data demonstrates beyond a reasonable doubt an increasingly more and more difficult role of UIA clipping. The trends, especially in Europe, are unswervingly changing in favor of endovascular treatment. Consequently, patients are qualified for microsurgery less frequently. However, the ones that are elected to undergo surgery commonly constitute the most complicated and difficult cases. This may result in relatively worse microsurgical outcomes now, as well as in the future.

Moreover, because of that strategy there were no hemorrhages in the UIAs under observation, which is a pertinent, most desired result. Additionally, we believe that it is worth underlining that AComA UIAs compromise a distinct, challenging group among all UIAs and for that reason require a detailed analysis performed here. We discuss our results in comparison to related work in section 4.1 through 4.3 of the paper. We would appreciate any pointers to additional works which should be considered in our discussion.

Finally, we would make the case that studies reporting UIA treatment results are needed in the literature so there will be material for future analyses. In other words, we contribute detailed descriptions to the pool of reported UIAs. This is important, because RCTs are, understandably, hard to design for UIAs.

Point 2: In addition, the 3 groups are controlled with different methods (CTA vs. DSA. vs. no regular follow-up during surgery).

Response 2: We thank the Reviewer for this comment. We agree with the Reviewer that each group underwent control studies that were tailored to a given management option. However, please note that in the microsurgical group (since 2018, when it became available in our program) we started routinely using intraoperative ICG after UIA clipping (as per subsection 2.1 “Follow–up algorithm” in the Materials and Methods section of our manuscript). We believe that initial DSA is necessary in most cases after an endovascular treatment, while a CTA is a good follow–up study after a microsurgery. In addition, we believe that exposing patients under observation to the potential risks of DSA does not justify the potential benefit of doing angiography in each case.

Finally, we would make a case that different methods of follow–up studies are fully supported by the recent literature, e.g., 2022 European Stroke Organisation (ESO) guidelines on management of unruptured intracranial aneurysms (quote: “We suggest that MR-angiography should be the primary tool for follow up imaging of endovascularly treated aneurysms, CTA for microsurgically treated aneurysms that DSA should only be considered if MRA and CTA are not conclusive”) or 2015 Guidelines for the Management of Patients With Unruptured Intracranial Aneurysms: A Guideline for Healthcare Professionals From the American Heart Association/American Stroke Association [quote: "Diagnosing/Imaging: Recommendations 1. DSA can be useful compared with noninvasive imaging for identification and evaluation of cerebral aneurysms if surgical or endovascular treatment is being considered (Class IIa; Level of Evidence B). 2. DSA is reasonable as the most sensitive imaging for follow-up of treated aneurysms (Class IIa; Level of Evidence C). 3. CTA and MRA are useful for detection and followup of UIA (Class I; Level of Evidence B). 4. It is reasonable to perform MRA as an alternative for follow-up for treated aneurysms, with DSA used as necessary when deciding on therapy (Class IIa; Level of Evidence C). 5. Coiled aneurysms, especially those with wider neck or dome diameters or those that have residual filling, should have follow-up evaluation (Class I; Level of Evidence B). The timing and duration of follow-up is uncertain, and additional investigation is necessary. 6. The importance of surveillance imaging after endovascular treatment of UIAs lacking high-risk features for recurrence remains unclear, but surveillance imaging is probably indicated (Class IIa; Level of Evidence C)”].

Point 3: The follow-up of endovascular treatment (DSA) is very short with an average of 7 months. Only just under 50 % were monitored over 24 months.

Response 3: It is indeed a good point to be made about our material, as we have described in the Material and Methods (subsection 2.1 “Follow–up algorithm”) as well as Results (subsection 3.2.3. Radiological outcomes) sections. We would like to highlight that our follow–up policy after endovascular treatment is in line with recent 2022 European Stroke Organisation (ESO) guidelines on management of unruptured intracranial aneurysms:

“PICO 5: In adult patients with occluded unruptured aneurysms, does any type and frequency of follow-up imaging compared to no follow-up imaging improve outcome (increase in QALYs)?

For adult patients with a treated UIA in whom aneurysm re-treatment remains an option, we suggest an initial radiological follow-up 3 to 12 months after UIA repair to detect potential UIA remnants or recurrence.”

Point 4: Radiological follow-up in the observation group was only available in 87.5%. Thus, a statement is also only possible in these patients.

Response 4: We would like to point out that clinical outcomes were available for 98.4% of patients in the observation group (as per subsection “3.1.3. Clinical outcomes”). We believe that clinical outcomes (e.g., SAH, mortality, morbidity) are much more important than radiological.

Point 5: The difference in numbers in Figure 2 (A vs. B) cannot be explained.

Response 5: It is our understanding and intention of Figure 2 that A and B would not be directly compared as they present different aneurysm shape descriptions – size ratio and aspect ratios, respectively. However, within these parameters, the differences between the observational and either treatment group were noticeable. As stated in the results (subsection 3.1.1. Assignment) the size ratio was significantly smaller in the observational group in comparison to the endovascular and microsurgery groups (both p<0.0001) – that is visualized with Figure 2A. Similarly, the aspect ratio was significantly higher in the endovascular and microsurgery groups than in the observational group (p<0.0001 and p=0.0002, respectively) – as per Figure 2B. We believe there is no contradiction between these two figures. We kindly ask the reviewer to respond if this explanation is sufficient or we need to make it clearer in the manuscript.

Alternatively, if the Reviewer wished to point out the differences between the number of occurrences in each of the groups [e.g., 44 in the (A) Endovascular vs 43 in the (B) Endovascular] then the reason for those minor differences is missing data. We would be happy to address this in the text upon request.

Point 5: The reduced rate of morbidity / mortality in observation is referred to a study from 2006. More recent studies were desirable.

Response 5: We thank the Reviewer for this comment. We completely agree that the literature on natural history of aneurysms is lacking. In this example, we were searching for annual rupture rates of AComA UIAs. As we already mentioned in these responses as well as in our manuscript, AComA UIAs may exhibit twice the risk of rupture than other UIAs. Because of that, to accurately evaluate the results of our strategy, we reviewed the literature in search of the natural history of AComA UIAs. Unfortunately, the authors were not able to find a more recent study of equal quality (i.e., meta–analysis). For that reason, we used the data from a 2006 study.

Point 6: Minor points: Table 1 and 2 are almost identical - one could be deleted.

Response 6: We agree with the Reviewer that this is a minor point and that Tables 1 and 2 present similar data. If that will be required from the Editor one will be deleted. However, as of now, we believe that the two present an almost equally important message (UIAs as qualified and as treated) and would prefer to keep both of them.

Point 7: In Table 1, the age in the surgery group is missing.

Response 7: We thank the Reviewer for pointing this out – this has been corrected.

Reviewer 2 Report

It is a retrospective and long-period clinical study concerning different management strategies for unruptured anterior communicating artery (AComA) aneurysms. The statistic analysis and English language were satisfied. Some questions occurred when reviewing and were needed to be solved.

Materials and methods

A specific description of the remodeled standards for choosing different strategies should be provided. The early patients using the algorithm should be reassessed with the new standard and the result needed to be compared with the primary group.

According to my knowledge, MRA would have a lower sensitivity for those who underwent microsurgery, and may cause a potential bias.

For the patients who underwent endovascular treatment, only receiving radiological examinations after 6 months was not enough for the high recurrent risk of endovascular treatment. There is also no related report of the recurrence in the present study which raises the reviewer’s concern. For the patients who underwent microsurgery, also no recurrence report was found.

Results

The rate of ischemic stroke after microsurgery was relatively high, according to Nussbaum et al., which was also quoted in the article, the rate seems to be only 2.1%.

Discussion

The authors are suggested to discuss more about how to choose more suitable strategies instead of repeating results and simply comparing with other research.

Quality of English language was satisfied.

Author Response

Response to Reviewer 2 Comments

It is a retrospective and long-period clinical study concerning different management strategies for unruptured anterior communicating artery (AComA) aneurysms. The statistic analysis and English language were satisfied. Some questions occurred when reviewing and were needed to be solved.

Point 1: Materials and methods A specific description of the remodeled standards for choosing different strategies should be provided. The early patients using the  algorithm should be reassessed with the new standard and the result needed to be compared with the primary group.

Response 1: We would like to thank the Reviewer for this suggestion. Undeniably, such analysis would be valuable and interesting. Unfortunately, we deem it impossible to perform as changes in our management strategy underwent development slowly over time – it was an evolution, not a revolution. For that reason, we cannot point a specific time with an “old” strategy happening before and the “new” after.

Point 2: According to my knowledge, MRA would have a lower sensitivity for those who underwent microsurgery, and may cause a potential bias.

Response 2: We agree with the Reviewer on that matter. However, please be aware that we had a different approach. As described in the Material and Methods, subsection “2.1. Follow-up algorithm” – after microsurgery we routinely waived routine post op vascular imaging (including MRA). Since 2018 we have been using intraoperative IC green videoangiography as confirmation.

Point 3: For the patients who underwent endovascular treatment, only receiving radiological examinations after 6 months was not enough for the high recurrent risk of endovascular treatment. There is also no related report of the recurrence in the present study which raises the reviewer’s concern. For the patients who underwent microsurgery, also no recurrence report was found.

Response 3: We thank the Reviewer for this comment. We agree that long–term follow–up after endovascular treatment is desired, but at the same time it comes with calculable costs. These are not hospital related expenses, but also risks and disadvantages for the patients. In this paper we are describing our strategy, which included, in general, a DSA at 6 months after endo treatment. If that showed a satisfactory result of the endo treatment, we ended our in-hospital surveillance (nevertheless, some of these patients had subsequent outpatient MRA follow-up). This is described in detail in the Materials and Methods, under subsection “2.1. Follow-up algorithm”.

With that in mind, please note that our follow–up policy after endovascular treatment is in line with recent 2022 European Stroke Organisation (ESO) guidelines on management of unruptured intracranial aneurysms:

“PICO 5: In adult patients with occluded unruptured aneurysms, does any type and frequency of follow-up imaging compared to no follow-up imaging improve outcome (increase in QALYs)?

For adult patients with a treated UIA in whom aneurysm re-treatment remains an option, we suggest an initial radiological follow-up 3 to 12 months after UIA repair to detect potential UIA remnants or recurrence.”

Point 4: Results The rate of ischemic stroke after microsurgery was relatively high, according to Nussbaum et al., which was also quoted in the article, the rate seems to be only 2.1%.

Response 4: We agree with the Reviewer that the rate of ischemic strokes after microsurgery is high in the reported material. The trends in Europe are unswervingly changing in favor of endovascular treatment. The above-mentioned complication rate is a mere reflection of this fact. We even have highlighted it in the conclusions with the phrase “the morbidity and mortality are compelling [..]”. Additionally, we have pointed this out in the Discussion in the subsection “4.3. Microsurgical group”.

Point 5: Discussion The authors are suggested to discuss more about how to choose more suitable strategies instead of repeating results and simply comparing with other research.

Response 5: We are grateful to the Reviewer for this suggestion. To address this, we have expanded subsection “4.4. Final comments” of our Discussion and added the following paragraph:

“Based on our experience, we conjecture evaluating each AComA UIA in a multidisci-plinary team, whenever possible and reasonable with a preference to observe. Prior to such evaluation a patient should be carefully evaluated using all available UIAs scales (PHASES, UIATS, ISUIA).”

We kindly ask the Reviewer to inform us if that will suffice.

Round 2

Reviewer 1 Report

All points have been adressed. No further comments or issues.